# Federated Graph Anomaly Detection via Disentangled Representation Learning

## Abstract

Graph anomaly detection plays a crucial role in identifying nodes that deviate significantly from normal patterns within a graph, with applications spanning various domains such as fraud detection, authorship fraud, and rumor propagation. Traditional methods primarily focus on aggregating information from neighboring nodes and reconstructing the central node based on these aggregated features. The anomaly degree is then calculated by comparing the reconstructed features with the original ones. Despite their effectiveness, these methods face limitations due to the constraints of device performance and the need to protect user privacy. In reality, graph data is often partitioned and distributed across different local clients, which leads to isolated client subgraphs. This partitioning results in incomplete feature aggregation, as the connections between subgraphs are missing, ultimately reducing the performance of anomaly detection models. To overcome these challenges, a federated graph anomaly detection approach based on disentangled representation learning is proposed. This method separates node features into two distinct components: intrinsic features and subgraph style features. By identifying outliers within the subgraph style features, a set of pseudo-nodes is generated and shared across the entire graph. These pseudo-nodes simulate connections between otherwise isolated subgraphs, which enables more comprehensive aggregation of intrinsic features from neighboring nodes. In addition, conditional variational autoencoders (CVAE) are employed alongside contrastive learning strategies to alleviate class imbalance and achieve effective feature disentanglement. These techniques help ensure that anomalous nodes are detected more accurately despite the inherent challenges of federated graph systems. Extensive experiments conducted on six diverse datasets provide compelling evidence of the proposed method's superior performance in federated graph anomaly detection, highlighting its ability to effectively handle incomplete graph structures while maintaining data privacy.

## CCS Concepts

• **Security and privacy** → *Web application security*; • **Computing methodologies** → **Neural networks**.

## Keywords

Disentangled Representation Learning, Federated Learning, Graph Anomaly Detection, Graph Neural Network

**ACM Reference Format:**

. 2018. Federated Graph Anomaly Detection via Disentangled Representation Learning. In *Proceedings of Make sure to enter the correct conference title from your rights confirmation emai (Conference acronym 'XX)*. ACM, New York, NY, USA, 9 pages. https://doi.org/XXXXXXX.XXXXXXX

## 1 Introduction

Graph anomaly detection aims to identify nodes in a graph that significantly deviate from features of most normal nodes, thereby uncovering potential risks and minimizing losses. There have been wide-ranging applications of this task, such as detecting false authorship in citation networks, fraudulent reviews in product review networks, and fraudulent accounts in transaction networks.

There have been methods designed for whole-graph anomaly detection, such as DAGAD [13] with data augmentation and class-wise losses. GAD-NR [22] with neighbor information reconstruction to handle attribute anomalies but also topological anomalies. The anomaly level of a node is assessed by discrepancies between its reconstructed features and original features. Relying on complete graph structure information, such methods enable effective data augmentation and neighborhood reconstruction, resulting in qualified detection performance.

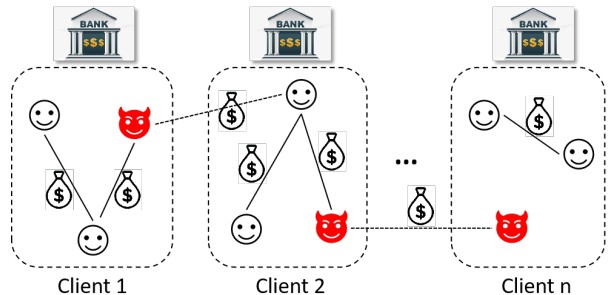

**Figure 1: A figure to illustrate the challenge of inter-client edges missing in graph anomaly detection.**

However, a single device may not be able to handle the entire graph, and privacy should not be leaked between different clients. Therefore, real-world graphs are partitioned and stored across different clients in some cases. Unlike tabular data, graph-structured data involves interconnections between samples. Storing the entire graph across separate clients means that connections between clients are absent, as shown in Figure 1. We evaluate the impact of the lack of edges between client subgraphs on the performance of graph anomaly detection, as shown in Figure 2(detailed experimental settings are illustrated in Section 5). Across six datasets, our local detection model shows significantly lower detection performance when client subgraphs are disconnected compared to when the global detection model is trained with complete graph information. Federated learning, which involves training models with data distributed across multiple local systems [16], effectively addresses

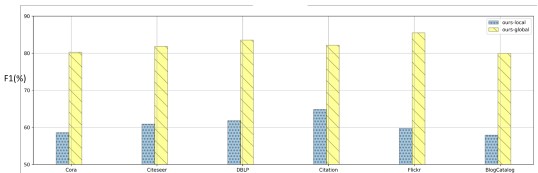

**Figure 2: Comparison of detection performance between the whole graph(ours-global) and the client subgraph(ours-local).**

the challenge of collaborative training across multiple clients and achieve client privacy protection. Compared to traditional single-model detection methods training on the whole-graph directly, federated graph anomaly detection lack aggregation of inter-client information. This reduces the richness of neighborhood aggregation along with the performance of the detection model. There have been methods tailored for federated graph learning, such as FedSage [31] generating missing cross-client neighbors with neighborhood distribution predictors and FedGTA [12] performing aggregation with local smoothing confidence as aggregation weights. However, such methods may perform suboptimally for anomaly detection due to the absence of specific designs for anomalies.

Inspired by disentangled representation learning [9], we propose a federated graph anomaly detection method to achieve cross-client neighborhood aggregation in graph anomaly detection. We disentangle node features into intrinsic features and subgraph style features. Intrinsic features are used for classification tasks of anomaly detection, while subgraph style features are used to construct globally shared nodes, simulating edges between clients. Specifically, this method comprises three main parts: (a) Local Autoencoders: With conditional variational autoencoders (CVAE) [23], this module derives the intrinsic features and subgraph style features for each node on each client according to feature and structure, respectively. (b) Local Feature Disentanglement: Leveraging contrastive learning strategies, this module generates negative node pairs with CVAE to achieve feature disentanglement and alleviate the classification imbalance problem in anomaly detection. (c) Global Shared Node Pool Construction: Within each client subgraph, this module identifies a few nodes likely to have connections outside the subgraph with subgraph style features. It then generates pseudo-features for these nodes with VAE out of privacy protection and shares them globally. After that, each node aggregates both local neighbors and simulated global neighbors, improving the model's detection performance.

The main contributions are summerized as follows:

- We propose a framework that disentangles node features into intrinsic features and subgraph style features to address graph anomaly and missing inter-client edges.
- We propose to adopt CVAE along with contrastive learning strategies to construct negative samples, ensuring intrinsic features and subgraph style features convey different meanings.
- We propose to separate out subgraph style features based on disentangled learning, constructing a globally shared node pool with a few selected nodes, to cope with the issue of missing edges between client subgraphs in federated graph anomaly detection.

- Experiments on real-world datasets demonstrate the superiority of our method in comparison with the state-of-the-art graph anomaly detection methods and graph federated learning methods.

## 2 Preliminaries

### 2.1 Hypothesis of Disentangled Learning

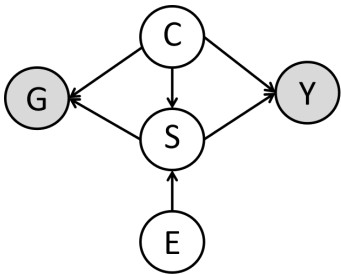

**Figure 3: Causal relationship diagram in federated graph anomaly scenarios.**

Taking a latent-variable model perspective on node features in federated graph anomaly detection, $Z \rightarrow Y$ is regarded as the mapping from features to labels, where $Z \subseteq R^n$ as the latent feature space, and $Y = \{0, 1\}^N$ as node labels. Following previous works [1, 5, 26], we partition the latent variable $Z$ into an invariant part $C$ and a varying part $S$, such that $Z = C + S$, depending on whether they are affected by $E$. In the context of anomaly detection, $C$ is understood as inherent features and used as features, indicating whether a node is anomalous, while $S$ represents subgraph style in respective clients, indicating connections within the subgraph. $E$ is regarded as the environment of each client. $G$ represents the overall graph. Thus, the relationship of variables $Z \rightarrow Y$ can be described in Figure 3.

### 2.2 Problem Definition

$G = \{V, E, X\}$ is defined as the imperceptible whole graph, where $V$ and $E$ are sets of nodes and edges, and $X$ is the node attribute matrix. In federated system, there would be a central server and $M$ clients owning respective subgraphs, namely client $i$ owns the subgraph $G_i = \{V_i, E_i, X_i\}$. Following vanilla algorithm of FedAvg, we concentrate on the scenario that there are no overlapping nodes across different clients and that edges across clients are absent in this federated system. As for graph anomaly detection, each node should be predicted as either 'normal' or 'abnormal', i.e., $V \rightarrow Y = \{0, 1\}^n$. The goal is to obtain a overall detection model on the server with parameters $\theta$ updated by training local client models according to the following formula:

$$\theta = \sum_{i=1}^{M} \frac{|V_i|}{\sum_{j=1}^{M} |V_j|} \theta_i \tag{1}$$

## 3 Proposed Method

In this section, we introduce the framework of our method. As shown in Figure 4, our method is based on an autoencoder structure. In a client, a local encoder-decoder model is designed to learn

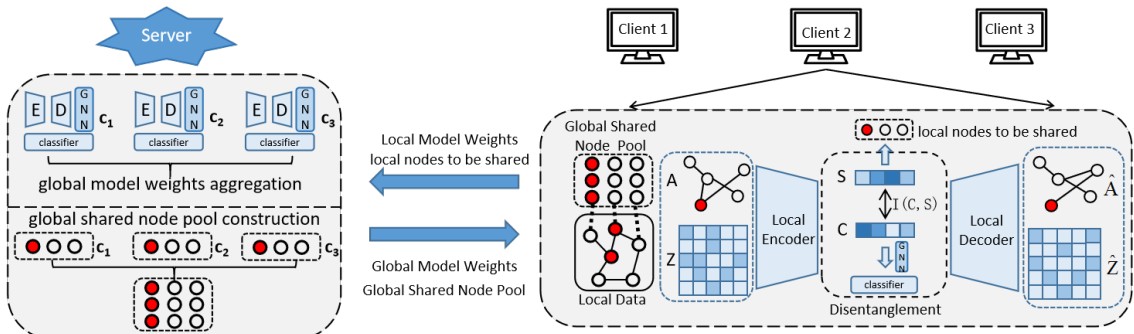

**Figure 4: The framework of our method.**

latent space features of nodes, generating two parts: $C$ and $S$ from the views of feature and structure. They are then disentangled with mutual information, which are approximated through contrastive learning, to represent intrinsic features and subgraph style features, respectively. Then local nodes to be shared are selected with subgraph style features. They are uploaded to the server along with the weights of the local model. On the server side, a global shared node pool is established to enrich neighborhood aggregation by simulating cross-client neighbors. Obtained by averaging local model weights, the global model weights are sent back to the clients for subsequent training along with the global shared node pool. In the following subsections, we will introduce in the order of local autoencoders, local feature disentanglement, global shared node pool construction, and prediction with global information.

### 3.1 Local Autoencoders

More details about our local detection model are shown in Figure 5. We first obtain the latent space representations of the nodes using the local model within the client subgraphs. Following [6], we utilize an autoencoder structure as base of our model to detect anomalous nodes by identifying feature deviations. Unlike previous methods that primarily use features learned from graph reconstruction to directly detect anomalies, instead, our approach leverages intrinsic features and subgraph style features obtained from the autoencoder to cope with anomaly detection in the following steps.

Specifically, $Z$ is first obtained with a fully-connected layer to map the original attribute matrix $X$ of the graph nodes into the dimension of the feature latent space. Following previous works [1, 5, 26], we obtain $C$ and $S$ as follows:

$$C = \text{relu}(WZ + b) \tag{2}$$

$$S = Z - C \tag{3}$$

where $W, b$ are learnable for a fully connected layer. Relu is the activation funtion.

The encoder maps nodes of the input graph into a probabilistic latent space. Typically, it is achieved with multi-layer perceptrons (MLPs). The encoder outputs two matrices: the mean ($\mu$) and the variance ($\sigma^2$), which define a Gaussian distribution for each node in the latent space. Such process can be expressed as following:

$$\mu = \text{MLP}_\mu(C) \tag{4}$$

$$\mu^Y = \text{MLP}_{\mu^Y}(Y) \tag{5}$$

$$\log\sigma = \text{MLP}_\sigma(C) \tag{6}$$

where $\mu^Y$ is the encoded mean of $Y$.

Then we employ the reparameterization trick [10] to rewrite $p_Z(C|Z) = p(\epsilon)$, where $C = \mu + \epsilon\sigma, \epsilon \sim N(0, I)$. From the point of intrinsic features, the variational approximate posterior of $C$ can be expressed as:

$$q(C|Z, A) = \prod_{i=1}^{N} q(c_m|Z, A) \tag{7}$$

$$q(c_m|Z, A) = N(z_m|\mu_m, \text{diag}(\sigma_m^2)) \tag{8}$$

To indicate the client subgraph style, we generate the reconstructed $\tilde{A}^i$ for client $i$. The structure decoder is given with an inner product between latent variables:

$$p(A|S) = \prod_{m=1}^{N} \prod_{n=1}^{N} p(A_{mn}^i|s_m, s_n) \tag{9}$$

$$\tilde{A}_{mn}^i = p(A_{mn}^i = 1|s_m, s_n) = \text{sigmoid}(s_m^T s_n) \tag{10}$$

where $\tilde{A}_{mn}^i$ are elements of reconstructed $A^i$.

At this point, the optimization objective of the conditional variational autoencoder (CVAE) that indicate $C$ and $S$ could be calculated as follows:

$$\begin{aligned}
\mathcal{L}_{\text{CVAE}} &= \mathbb{E}_{q(S|Z,A)}[\log p(A|Z)] - \text{KL}[q(C|Z,A)||p(Z)] \\
&= -\frac{1}{N^2} \sum_{i=1}^{N} \sum_{j=1}^{N} A_{ij} \cdot \log(\tilde{A}_{ij}) + (1 - A_{ij}) \cdot \log(1 - \tilde{A}_{ij}) \\
&+ \frac{1}{2} \sum_{i=1}^{n} [(\mu_i - \mu_i^Y)^2 + \sigma_i^2 - \log\sigma_i^2 - 1]
\end{aligned} \tag{11}$$

where $\mathcal{L}_{\text{CVAE}}$ represents the CVAE loss.

### 3.2 Local Feature Disentanglement

As shown in Figure 5, $C$ and $S$ from the views of feature and structure have been generated with a local CVAE in the last subsection. Since $C$ will serve as the intrinsic feature for neighbor aggregation to detect anomalies, while $S$ will be used to express the local subgraph style and build the global shared node pool. In this subsection, we propose to minimize the mutual information between $C$ and $S$ to reduce the correlation between the two feature spaces,

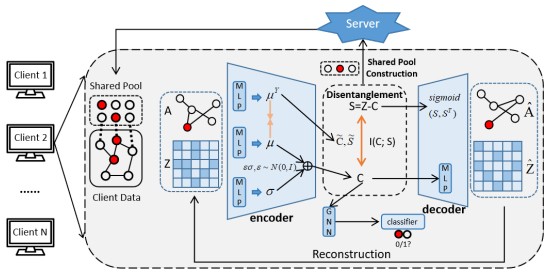

**Figure 5: The details about our local detection model.**

which means the necessity to derive an upper bound for $I(C; S)$ and minimize it. However, an exact estimate of $I(C; S)$ could be highly expensive [2, 18]. There has been considerable research on estimating lower bounds of mutual information [21]. Among them, contrastive learning sampling provides a practical approach for approximation [5]. Following the idea of contrastive learning sampling, we propose that an adjusted InfoNCE [18] could be capable of estimating an upper bound of mutual information.

To facilitate explanation, we first outline the form of InfoNCE used in the anomaly detection task described in this paper, which could be regarded as an estimate of the lower bound of $I(C; S)$.

$$I(C; S) \geq \mathcal{L}_{\text{infoNCE}}$$
$$= \mathbb{E}_{\{c,s\}\sim P_{pos},\{c_i,s\}\sim P_{neg}} \log \frac{e^{\phi(c,s)}}{e^{\phi(c,s)} + \sum_i e^{\phi(c_i,s)}} \quad (12)$$

where $P_{pos}$ means the distribution of positive sample pairs between $C$ and $S$, while $P_{neg}$ means the distribution of negative sample pairs. The function $\phi(c, s)$ measures the similarity between $c$ and $s$.

Since $p(c|s)$ could be expressed by similarity functions [18]:

$$e^{\phi(c,s)} \propto \frac{p(c|s)}{p(c)} \quad (13)$$

Similarly, we can derive the upper bound of $I(C; S)$.

$$I(C; S) - \log(N) = -\mathbb{E}_C \log[\frac{p(c)}{p(c|s)}N]$$
$$\leq -\mathbb{E}_C \log[\frac{p(c)}{p(c|s)}(N-1)]$$
$$= -\mathbb{E}_C \log[\frac{p(c)}{p(c|s)}(N-1)\mathbb{E}_{c_i}\frac{p(c_i|s)}{p(c_i)}]$$
$$\approx -\mathbb{E}_C \log[\frac{p(c)}{p(c|s)}(N-1)\sum_{c_i \in C^{neg}}\frac{p(c_i|s)}{p(c_i)}] \quad (14)$$
$$= \mathbb{E}_C \log[\frac{\frac{p(c|s)}{p(c)}}{\sum_{c^i \in C^{neg}}\frac{p(c_i|s)}{p(c_i)}}]$$
$$= \mathbb{E}_{\{c,s\}\sim P_{pos},\{\hat{c},\hat{s}\}\sim P_{neg}} \log \frac{e^{\phi(c,s)}}{\sum e^{\phi(\hat{c},\hat{s})}}$$

In practice, we define positive sample pairs $(c, s)$ as two feature parts from the same node and design negative sample pairs $(\hat{c}, \hat{s})$ in Eq.(14) as following:

$$\widetilde{c} = (1 - \mu^Y) + \epsilon\sigma \quad (15)$$

$$\widetilde{s} = z - \widetilde{c} \quad (16)$$
$$(\hat{c}, \hat{s}) \in \{(\widetilde{c}, s), (c, \widetilde{s})\} \quad (17)$$

Leveraging advantages of CVAE, we generate negative samples for contrastive sampling, which not only achieves the disentanglement of $C$ and $S$, but also make up the rarity of anomaly information to alleviate the class imbalance problem in anomaly detection. Therefore, the disentanglement loss could be expressed as:

$$\mathcal{L}_D = \mathbb{E}_{p(c,s)}\log \frac{e^{\phi(c,s)}}{e^{\phi(\widetilde{c},s)} + e^{\phi(c,\widetilde{s})}} \quad (18)$$

where $\mathcal{L}_D$ is the disentanglement loss.

### 3.3 Global Shared Node Pool Construction

Through the aforementioned two steps, we obtain the local feature space $S$ that independently represents the structure styles of client subgraphs. In this section, we will construct a shared node pool with $S$ uploaded from all clients to simulate the connections across client subgraphs.

To ensure balanced distribution across the entire graph, we sample the same proportion (e.g., 10%) of pseudo-nodes from each client and place them into the shared node pool. For privacy protection, we use pseudo-node features rather than original node features as globally shared features. The selection process for pseudo-nodes is as follows:

$$\overline{S}_i = \text{Mean}(S_i) \quad (19)$$
$$V_i^{sampled} = \text{Sort}(S_i, \overline{S}_i) \quad (20)$$

where $\text{Mean}(S_i)$ calculates the mean of $S_i$, and $V_i^{sampled}$ samples a proportion of nodes sorted with the cosine similarity between $s_i$ and $\overline{S}_i$. According to the consistency assumption of GNN [33] that neighboring nodes have similar features, it's naturally infer that the node which is most dissimilar to $\text{Mean}(S_i)$ is more likely to have connections with nodes in other clients.

Then pseudo intrinsic features of shared nodes of each client could be expressed as:

$$C_i^{shared} = \mu + \epsilon'\sigma, \epsilon' \sim N(0, I) \quad (21)$$
$$C^{shared} = \text{Concat}(\{C_i^{shared}\}) \quad (22)$$

Starting from the second round of federated learning, each client's subgraph space during training incorporates simulated cross-client information. This is reflected in the enlarged node feature matrix and the expanded adjacency matrix. Taking the training process on client $i$ as an example, we first obtain the subgraph style features of the shared pools as follows:

$$Z^{shared} = \text{MLP}_{decoder}(C^{shared}) \quad (23)$$
$$S^{shared} = Z^{shared} - C^{shared} \quad (24)$$

Thus, based on the new enlarged subgraph style feature matrix $\hat{S}_i$, we calculate the enlarged adjacency matrix $\hat{A}^i$.

$$\hat{S}_i = \text{Concat}(S_i, S^{shared}) \quad (25)$$

$$\hat{A}_{mn}^i = \begin{cases} 1, & \text{if } A_{mn}^i = 1 \text{ and } v_m, v_n \in V_i \\ \text{sigmoid}(s_m^T s_n), & \text{if } v_m \in V_i \text{ xor } v_n \in V_i \\ 0, & \text{otherwise} \end{cases} \quad (26)$$

where $\hat{A}^i$ includes edges to simulated nodes outside the client $i$.

| Methods | Client Memory | Server Memory | Client Time | Server Time |
|---|---|---|---|---|
| FedAvg | $O((b+k)f+f^2)$ | $O(N+Nf^2)$ | $O(kmf+nf^2)$ | $O(N)$ |
| FedProx | $O((b+k)f+2f^2)$ | $O(N+Nf^2)$ | $O(kmf+nf^2+f^2)$ | $O(N)$ |
| FedSage+ | $O(L(n+sg)f+3Lf^2)$ | $O(N+3Nf^2)$ | $O(L(m+sg)f+L(n+sg)f^2)$ | $O(N)$ |
| FedEgo | $O(N(b+k)f+f^2)$ | $O(N^2+Nf^2)$ | $O(Nkmf+nf^2)$ | $O(N)$ |
| FedGTA | $O((b+k)f+f^2+kKc)$ | $O(N+Nf^2+NkKc)$ | $O(km(f+knc)+n(f^2+c))$ | $O(N+NkKc)$ |
| Ours | $O(2(b(1+\theta N)+k)f+3f^2)$ | $O(2N+3Nf^2)$ | $O(2kmf+3nf^2)$ | $O(N)$ |

Table 1: Complexity analysis of baseline federated methods. Let $n$, $m$, $c$, and $f$ denote the number of nodes, edges, classes, and feature dimensions, respectively. $s$ refers to the count of selected augmented nodes, while $g$ represents the number of generated neighbors. The batch size is indicated by $b$, and $T$ refers to the number of dynamic training rounds. $k$ signifies the number of feature aggregation steps, and $K$ represents the moment order. Furthermore, $N$ corresponds to the number of clients.

## 3.4 Prediction with Global Information

Following the previous steps, we obtain the global model parameters aggregated according to Eq. (1) from the server and construct the global shared node pool with nodes to be shared from each client. In this subsection, we first obtain the node feature $H_i$ of client $i$ involving information of shared global nodes for inputting the classifier and the cross entropy loss ($\mathcal{L}_{ce}$) function.

$$\hat{C}_i = \text{Concat}(C_i, C^{shared}) \quad (27)$$

$$H_i = \text{Concat}(\hat{C}_i, \text{Agg}(\hat{C}_i, \hat{A}_i)) \quad (28)$$

$$\mathcal{L}_{ce} = -\frac{1}{N}\sum_{i=1}^{N}[y_i\log(p_i(y_i)) + (1-y_i)\log(1-p_i(y_i))] \quad (29)$$

where Eq.(28) indcates the typical GNN message passing operation that combines the node's self-feature and the aggregation of its neighborhood. $p_i(y_i)$ is the predicted probability of the node $v_i$ on the class label $y_i$.

Therefore, the overall loss of our model comprises three components: the supervised classification loss $\mathcal{L}_{ce}$, the CVAE loss $\mathcal{L}_{CVAE}$, and the feature disentanglement loss $\mathcal{L}_D$. Thus, the total loss formula is as follows:

$$\mathcal{L} = \mathcal{L}_{ce} + \alpha \cdot \mathcal{L}_{CVAE} + \beta \cdot \mathcal{L}_D \quad (30)$$

where $\alpha$ and $\beta$ serve as hyperparameters.

## 3.5 Complexity Analysis

Following FedGTA [12], we conducted a complexity analysis, as shown in Table 1. To further clarify, we present the algorithmic complexity of each method in Table 1. For a $k$-layer GNN model with batch size $b$, the precomputed results are limited by a space complexity of $O((b+k)f)$. The overhead for linear regression is $O(f^2)$. For our proposed method, since the features are divided into two parts and a proportion $\theta$ of globally shared nodes is used, the memory overhead for storing node features is $O(2(b(1+\theta N)+k)f)$. The use of CVAE increases the number of model parameters to three times that of a single detection model. Thus, the space complexity on the client side is $O(2(b(1+\theta N)+k)f+3f^2)$. The complexity analysis for other components follows a similar approach. Given that our detection method processes the graph in batches rather than using the entire graph as input, and the number of globally shared nodes is relatively small, our approach maintains low time overhead while also demonstrating spatial scalability.

## 3.6 Analysis of Privacy Protection

In this method, to simulate the edges between client subgraphs, we construct a globally shared node pool, which is derived from information uploaded by each client. However, this does not involve privacy leakage, as the small number of node features uploaded by each client are not the original attributes but rather latent space features. These features are sampled from a Gaussian distribution and generated using a VAE, making them essentially different from the actual features of any given node.

## 4 Experiments

## 4.1 Experiment Setup

In this section, we conducted comparative experiments on six real-world datasets with five currently representative federated learning methods and three global graph anomaly detection methods to test the performance of our proposed approach. Additionally, we conducted ablation experiments to test the effectiveness of the designed modules. These contents were aimed at addressing the following questions:

**Q1**: Does our method outperform the state-of-the-art approaches in federated scenarios and achieve qualified detection performance in global (whole-graph) scenarios?

**Q2**: Can the incorporation of the feature disentanglement and the shared node pool lead to an improvement in federated detection performance?

**Q3**: Is our method sensitive to the key hyperparameters?

**Q4**: Can the feature disentanglement achieve the desired effect as anticipated?

| Dataset | Cora | Citeseer | DBLP | Citation | Flickr | BlogCatalog |
|---|---|---|---|---|---|---|
| Nodes | 2,708 | 3,327 | 5,484 | 8,935 | 7,575 | 5,196 |
| Edges | 5,429 | 4,732 | 8,117 | 15,098 | 241,277 | 172,759 |
| Features | 1,433 | 3,703 | 6,775 | 6,775 | 12,047 | 8,189 |
| Anomalies(%) | 5.53 | 4.51 | 4.98 | 4.86 | 5.94 | 5.77 |

Table 2: The characteristics of the datasets.

*4.1.1 Datasets.* We employ six datasets to evaluate the effectiveness of our method, with their statistical characteristics presented in Table 2. These datasets are real-world attributed graphs with injected anomalies following [14]. Each dataset in our experiments

| Dataset | Cora | | | | Citeseer | | | | DBLP | | | |
|---|---|---|---|---|---|---|---|---|---|---|---|---|
| Methods | Acc.(%) | Precision(%) | Recall(%) | F1(%) | Acc.(%) | Precision(%) | Recall(%) | F1(%) | Acc.(%) | Precision(%) | Recall(%) | F1(%) |
| ours-local (lower bound) | $94.58_{\pm0.28}$ | $58.78_{\pm2.40}$ | $58.79_{\pm0.78}$ | $58.57_{\pm1.36}$ | $95.77_{\pm0.60}$ | $68.23_{\pm2.96}$ | $57.37_{\pm0.84}$ | $60.87_{\pm0.65}$ | $95.63_{\pm0.33}$ | $85.74_{\pm3.41}$ | $57.70_{\pm4.16}$ | $61.79_{\pm5.91}$ |
| FedAvg(GCN) | $94.48_{\pm0.00}$ | $47.24_{\pm0.00}$ | $50.00_{\pm0.00}$ | $48.58_{\pm0.00}$ | $94.82_{\pm0.48}$ | $68.67_{\pm5.27}$ | $60.18_{\pm4.17}$ | $62.91_{\pm4.85}$ | $95.30_{\pm0.14}$ | $77.30_{\pm1.66}$ | $63.72_{\pm3.04}$ | $67.75_{\pm2.76}$ |
| FedProx(GCN) | $94.48_{\pm0.00}$ | $47.24_{\pm0.00}$ | $50.00_{\pm0.00}$ | $48.58_{\pm0.00}$ | $94.92_{\pm0.00}$ | $47.60_{\pm0.00}$ | $49.84_{\pm0.00}$ | $48.70_{\pm0.00}$ | $95.30_{\pm0.34}$ | $78.96_{\pm5.53}$ | $58.82_{\pm5.51}$ | $62.43_{\pm7.76}$ |
| FedSage+ | $95.07_{\pm0.29}$ | $63.84_{\pm14.49}$ | $56.83_{\pm5.94}$ | $58.66_{\pm8.65}$ | $95.71_{\pm0.97}$ | $76.00_{\pm1.11}$ | $64.34_{\pm7.05}$ | $68.18_{\pm8.38}$ | $95.28_{\pm0.41}$ | $74.81_{\pm2.90}$ | $63.55_{\pm0.98}$ | $67.20_{\pm0.70}$ |
| FedEgo | $93.92_{\pm0.30}$ | $66.11_{\pm9.69}$ | $52.18_{\pm1.26}$ | $52.73_{\pm2.21}$ | $94.78_{\pm0.34}$ | $60.25_{\pm6.17}$ | $51.73_{\pm2.52}$ | $52.07_{\pm4.32}$ | $95.34_{\pm0.25}$ | $74.21_{\pm6.49}$ | $55.37_{\pm0.54}$ | $58.06_{\pm1.05}$ |
| FedGTA | $91.78_{\pm1.29}$ | $60.87_{\pm4.47}$ | $64.37_{\pm2.79}$ | $61.30_{\pm3.04}$ | $94.39_{\pm0.31}$ | $66.68_{\pm1.85}$ | $64.25_{\pm1.95}$ | $63.17_{\pm3.86}$ | $96.17_{\pm0.78}$ | $79.56_{\pm4.01}$ | $81.10_{\pm2.07}$ | $78.37_{\pm2.06}$ |
| ours | $\mathbf{95.15}_{\pm0.28}$ | $\mathbf{77.42}_{\pm1.71}$ | $71.28_{\pm2.47}$ | $\mathbf{73.84}_{\pm1.91}$ | $\mathbf{95.72}_{\pm0.22}$ | $76.67_{\pm1.51}$ | $74.99_{\pm4.55}$ | $75.62_{\pm2.46}$ | $96.81_{\pm0.24}$ | $82.97_{\pm1.64}$ | $85.65_{\pm0.72}$ | $84.21_{\pm0.61}$ |
| DAGAD(GCN) | $93.36_{\pm1.15}$ | $70.30_{\pm3.07}$ | $76.40_{\pm2.85}$ | $72.75_{\pm2.90}$ | $94.56_{\pm0.98}$ | $70.07_{\pm4.48}$ | $74.28_{\pm3.78}$ | $71.81_{\pm3.87}$ | $94.21_{\pm0.41}$ | $71.02_{\pm1.45}$ | $75.15_{\pm3.07}$ | $72.77_{\pm1.70}$ |
| DAGAD(GAT) | $94.54_{\pm0.82}$ | $74.01_{\pm3.51}$ | $76.71_{\pm4.25}$ | $75.26_{\pm3.81}$ | $95.46_{\pm1.03}$ | $74.02_{\pm5.80}$ | $76.34_{\pm5.06}$ | $75.07_{\pm5.35}$ | $94.68_{\pm1.34}$ | $73.58_{\pm5.23}$ | $78.10_{\pm3.95}$ | $75.47_{\pm4.54}$ |
| GAD-NR | $95.50_{\pm0.86}$ | $79.25_{\pm0.61}$ | $74.61_{\pm1.38}$ | $76.69_{\pm1.67}$ | $96.61_{\pm0.15}$ | $82.13_{\pm1.53}$ | $74.51_{\pm1.13}$ | $77.73_{\pm0.51}$ | $96.42_{\pm0.25}$ | $83.69_{\pm2.45}$ | $74.34_{\pm1.30}$ | $78.14_{\pm0.91}$ |
| ours-global (upper bound) | $96.25_{\pm0.64}$ | $84.38_{\pm4.81}$ | $77.61_{\pm4.20}$ | $80.25_{\pm3.02}$ | $96.89_{\pm0.57}$ | $82.55_{\pm4.35}$ | $81.43_{\pm2.45}$ | $81.82_{\pm2.35}$ | $97.42_{\pm0.18}$ | $91.14_{\pm0.79}$ | $77.23_{\pm1.33}$ | $83.52_{\pm1.20}$ |

| Dataset | Citation | | | | Flickr | | | | BlogCatalog | | | |
|---|---|---|---|---|---|---|---|---|---|---|---|---|
| Methods | Acc.(%) | Precision(%) | Recall(%) | F1(%) | Acc.(%) | Precision(%) | Recall(%) | F1(%) | Acc.(%) | Precision(%) | Recall(%) | F1(%) |
| ours-local (lower bound) | $95.56_{\pm0.71}$ | $85.11_{\pm1.26}$ | $61.41_{\pm0.85}$ | $64.81_{\pm0.80}$ | $91.66_{\pm3.88}$ | $75.22_{\pm0.83}$ | $59.74_{\pm0.58}$ | $59.78_{\pm1.32}$ | $92.53_{\pm0.52}$ | $62.24_{\pm6.26}$ | $56.59_{\pm3.36}$ | $57.92_{\pm3.73}$ |
| FedAvg(GCN) | $95.73_{\pm0.71}$ | $81.65_{\pm1.33}$ | $64.78_{\pm1.32}$ | $69.73_{\pm1.11}$ | $93.94_{\pm0.00}$ | $46.97_{\pm0.00}$ | $50.00_{\pm0.00}$ | $48.44_{\pm0.00}$ | $94.15_{\pm0.00}$ | $47.08_{\pm0.00}$ | $50.00_{\pm0.00}$ | $48.49_{\pm0.00}$ |
| FedProx(GCN) | $95.16_{\pm0.89}$ | $77.99_{\pm2.69}$ | $53.96_{\pm1.73}$ | $55.90_{\pm2.77}$ | $93.94_{\pm0.00}$ | $46.97_{\pm0.00}$ | $50.00_{\pm0.00}$ | $48.44_{\pm0.00}$ | $94.15_{\pm0.00}$ | $47.08_{\pm0.00}$ | $50.00_{\pm0.00}$ | $48.49_{\pm0.00}$ |
| FedSage+ | $95.37_{\pm0.50}$ | $72.31_{\pm3.75}$ | $67.98_{\pm4.57}$ | $69.85_{\pm4.28}$ | $94.08_{\pm0.95}$ | $73.38_{\pm2.99}$ | $70.37_{\pm1.86}$ | $71.71_{\pm0.97}$ | $93.85_{\pm0.82}$ | $71.09_{\pm0.36}$ | $65.43_{\pm2.95}$ | $67.61_{\pm2.04}$ |
| FedEgo | $95.26_{\pm0.65}$ | $79.61_{\pm0.62}$ | $55.48_{\pm2.03}$ | $58.26_{\pm2.94}$ | $94.06_{\pm0.63}$ | $72.77_{\pm5.91}$ | $60.51_{\pm3.37}$ | $63.83_{\pm4.33}$ | $93.67_{\pm0.23}$ | $75.46_{\pm2.30}$ | $59.45_{\pm1.76}$ | $63.08_{\pm2.28}$ |
| FedGTA | $95.81_{\pm0.27}$ | $76.32_{\pm1.95}$ | $78.92_{\pm1.59}$ | $77.12_{\pm1.79}$ | $93.82_{\pm0.50}$ | $84.98_{\pm1.56}$ | $68.14_{\pm3.83}$ | $73.20_{\pm3.63}$ | $93.87_{\pm0.11}$ | $71.14_{\pm3.22}$ | $63.49_{\pm2.31}$ | $66.19_{\pm2.76}$ |
| ours | $\mathbf{96.87}_{\pm0.25}$ | $\mathbf{85.79}_{\pm1.81}$ | $79.06_{\pm3.59}$ | $\mathbf{81.87}_{\pm2.40}$ | $\mathbf{95.30}_{\pm0.37}$ | $82.41_{\pm4.97}$ | $71.74_{\pm2.15}$ | $75.57_{\pm0.45}$ | $94.59_{\pm0.47}$ | $76.17_{\pm3.15}$ | $67.92_{\pm0.25}$ | $\mathbf{71.09}_{\pm1.18}$ |
| DAGAD(GCN) | $94.62_{\pm0.41}$ | $72.22_{\pm1.51}$ | $78.42_{\pm2.43}$ | $74.82_{\pm1.54}$ | $95.27_{\pm1.92}$ | $82.39_{\pm2.81}$ | $79.48_{\pm5.27}$ | $79.49_{\pm3.04}$ | $95.57_{\pm1.79}$ | $83.11_{\pm7.37}$ | $78.56_{\pm5.02}$ | $79.54_{\pm3.50}$ |
| DAGAD(GAT) | $95.17_{\pm0.67}$ | $74.58_{\pm2.84}$ | $79.03_{\pm1.24}$ | $76.53_{\pm2.12}$ | $93.63_{\pm0.67}$ | $72.08_{\pm2.86}$ | $76.84_{\pm5.28}$ | $74.12_{\pm3.12}$ | $91.59_{\pm3.68}$ | $67.41_{\pm6.36}$ | $71.13_{\pm4.77}$ | $68.41_{\pm5.87}$ |
| GAD-NR | $95.59_{\pm0.38}$ | $76.51_{\pm2.44}$ | $73.60_{\pm0.15}$ | $74.92_{\pm1.15}$ | $88.89_{\pm0.00}$ | $51.80_{\pm0.00}$ | $51.96_{\pm0.00}$ | $51.87_{\pm0.00}$ | $87.45_{\pm0.00}$ | $51.26_{\pm0.00}$ | $51.74_{\pm0.00}$ | $51.35_{\pm0.00}$ |
| ours-global (upper bound) | $96.92_{\pm0.17}$ | $85.26_{\pm1.22}$ | $79.70_{\pm2.77}$ | $82.14_{\pm1.58}$ | $97.00_{\pm0.14}$ | $88.32_{\pm1.04}$ | $83.13_{\pm0.93}$ | $85.50_{\pm0.64}$ | $95.38_{\pm0.16}$ | $78.82_{\pm0.79}$ | $81.15_{\pm0.41}$ | $79.93_{\pm0.49}$ |

**Table 3: Performance comparison of our method on six datasets.**

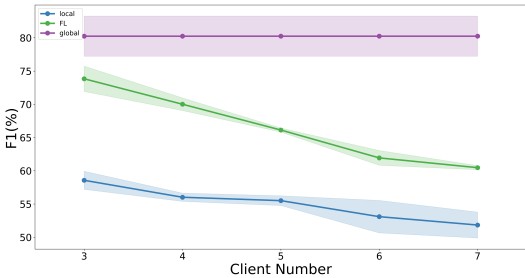

**Figure 6: Performance comparison of our method on Cora under different number of clients.**

is split into three parts: 60% for training, 20% for validation, and 20% for testing.

*4.1.2 Compared Methods.* Due to the current scarcity of research on federated graph anomaly detection, we test our method by comparing it with five representative and up-to-date federated graph learning models. Among them, FedAvg and FedProx are representative federated learning models. FedSage+, FedEgo, and FedGTA

are proposed recently. To validate our model in the global (whole-graph) scenario, we also select three recently proposed graph anomaly detection models, including DAGAD-GCN, DAGAD-GAT, and GAD-NR. Given the limited availability of supervised methods for the corresponding datasets, we extract the embeddings trained by GAD-NR and apply a linear classifier for supervised training.

*4.1.3 Experiment Settings and Implementation.* The learning rate is set as 0.002, and the weight decay rate is 1e-5. We set the embedding size to 64, the round number to 50, and training epochs in a round to 5. Following FedSage [31], we partition the whole-graph into several parts to simulate federated scenarios with Louvain.

Our model is implemented in PyTorch 1.13.1 [19] and PyG [8] with Python 3.9, while baselines are implemented with codes published by their authors on a single NVIDIA RTX 4090 GPU.

*4.1.4 Evaluation metrics.* We evaluate the effectiveness of our method with the following four metrics, accuracy, macro-precision, macro-recall, and macro-f1. Accuracy is a commonly used metric for classification tasks, while the others reflect the performance of models on imbalanced data without overly underestimating the minority (anomaly) class.

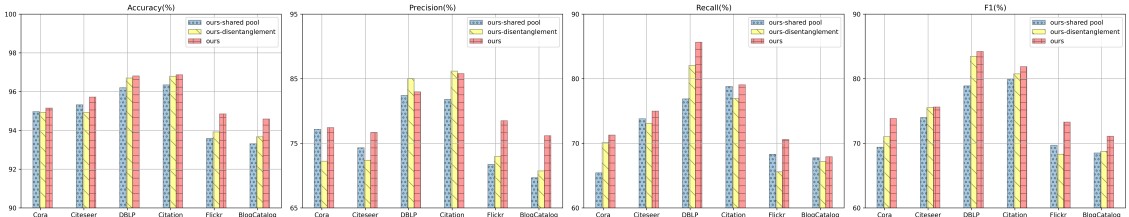

Figure 7: Ablation experiments on six datasets.

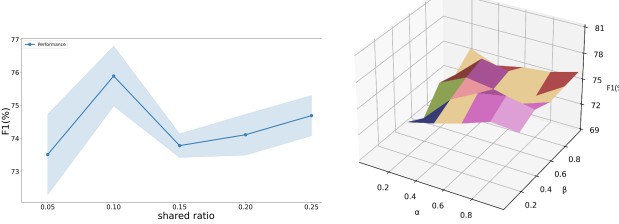

Figure 8: Experiments of sensitivity to hyper-parameters.

## 4.2 Overall Results (RQ1)

In our comparative experiments, we first conducted experiments under the setting of three clients, covering local, federated, and global scenarios. We recorded the experimental results as shown in Table 3. In the local scenario, our method involves independent training on each client without interaction with the server, representing the lower bound of federated graph anomaly detection methods. The global scenario entails detection on the entire graph, representing the upper bound of federated graph anomaly detection methods. We compare our method in the global scenario with three recent graph anomaly detection methods, DAGAD-GCN, DAGAD-GAT and GAD-NR. Our method, after disentangling features into intrinsic and style features and learning features from both attribute and structural perspectives, achieves superior detection performance.

In the context of federated graph learning, we first implemented classic FedAvg and FedProx combined with GCN as two baseline models. These models simply upload client model weights, perform weighted updates on the server, and include regularization terms. However, they only showed promising results on the DBLP and Citation datasets. Subsequently, we compared our method with other recent federated graph learning models. It's notable that due to the rarity of anomaly classes in anomaly detection, the difference in accuracy is minimal. Therefore, we focused on observing the Macro-F1 score, which comprehensively evaluates the classification performance of anomaly detection. FedSage+ relies solely on nodes themselves to generate the number and distribution of neighbors, while the quality of generated neighbors may be suboptimal. FedEgo shares the aggregated feature of the entire neighborhood, however, this approach may submerge the distinctive features of anomalous nodes. Among all baseline models, FedGTA performs the best. Nevertheless, due to the lack of specific design for graph anomaly detection, it fails to outperform our proposed method.

Under the setting of different number of clients, we conducted related experiments as shown in Figure 6. Observably, as the number

of clients increases, the missing connections between clients also increase, inevitably leading to a decline in model performance, which aligns with the trend observed in detection performance of the local scenario.

## 4.3 Ablation Study (RQ2)

To evaluate the effectiveness of two key modules of our method, we conducted series of ablation experiments under the setting of three clients by excluding each module, as shown in Figure 7, where ours-shared pool excludes the shared node pool for federated graph learning, and ours-disentanglement excludes the local feature disentanglement for decoupling intrinsic features and client subgraph style features. The shared pool plays a useful role in improving the performance of detection on all six datasets, while the introduction of the feature disentanglement module also benefits detection performance in most cases.

## 4.4 Sensitivity to Hyper-Parameters (RQ3)

**The ratio of the shared node pool.** According to the process of constructing the shared node pool, the ratio of nodes sampled in each client as shared nodes is a critical hyper-parameter. Hence, we evaluate our model's sensitivity to it on the Cora dataset, as shown in Figure 8. The model achieves the best performance when $\theta$ is around 10%.

**Weight factor analysis of the loss function.** As shown in Eq.(30), the two factors $\alpha$ and $\beta$ are critical in balancing the supervised classification loss, the disentanglement loss, and the CVAE loss. Hence, we evaluate our model's sensitivity to these terms under different settings.

Specifically, we run our method on the Cora dataset for $\alpha, \beta \in \{0.2, 0.4, 0.6, 0.8, 1.0\}$ based on grid search and report the results of all three evaluation metrics in Figure 8. It can be observed that our method is not sensitive to the two loss weight parameters $\alpha$ and $\beta$ in terms of macro F1. In particular, our method achieves better macro-F1 when $\alpha$ and $\beta$ are around 0.4 and 0.8, respectively.

## 4.5 Visualization Analysis (RQ4)

T-distributed stochastic neighbor embedding (TSNE) [25] is a feature visualization tool that can transform high-dimensional features into two-dimensional features to facilitate observation of the pattern of feature distribution. To evaluate whether the local feature disentanglement module achieves the expected effect, we utilize TSNE on all six datasets. Figure 9 shows the feature distribution after being handled with TSNE. It can be observed that the subgraph style feature $S$ of normal and abnormal nodes are more coupled,

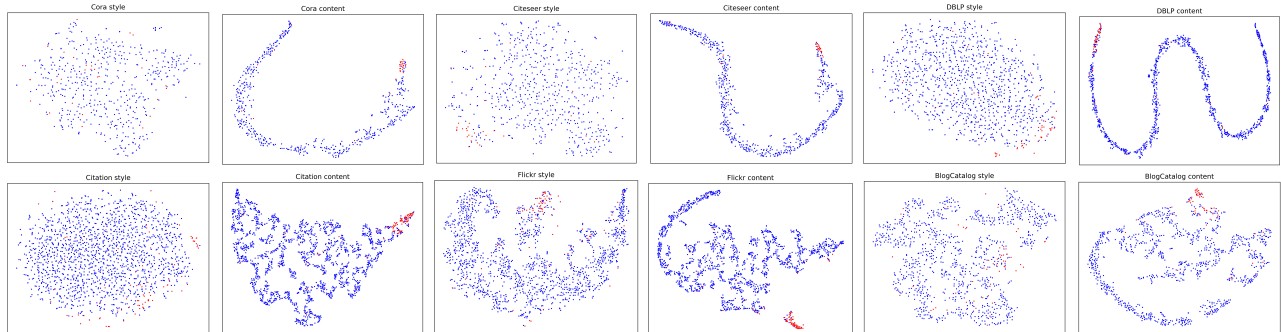

**Figure 9: The visualization of disentangled representation with TSNE on all six datasets.**

while the intrinsic feature $C$ present a more distinguishable border between normal and abnormal nodes. This demonstrates that the disentangled representation learning successfully distinguishes between features with different semantic meanings.

## 5 Related Work

### 5.1 GNN-based Graph Anomaly Detection

Graph Neural Networks (GNNs) are capable of generating high-quality node embeddings by aggregating information from neighboring nodes. Numerous studies have successfully applied GNN-based methods to detect anomalous nodes in diverse domains, such as citation networks [3], product review networks [15], social networks [17], and transaction networks [24], demonstrating the versatility and effectiveness of GNNs in tackling these problems.

GraphConsis [15] defined inconsistency of feature and context in graph anomaly, and conducted experiments on fraud comments datasets, followed with [7, 29]. CoLA [14] exploited local information by sampling contrastive instance pairs, which can capture the relationship between each node and its neighborhood. To better handle the high-frequency features of fraudsters, BWGNN [24] implemented filters of multiple frequency bands based on the Beta kernel designations. CONAD [27] first modeled prior human knowledge through a novel data augmentation strategy, then integrated the modeled knowledge in a Siamese graph neural network encoder through a contrastive loss. DAGAD [13] derived additional samples to enrich the training set and adopts class-wise losses to reveal the differences between anomalous and normal nodes with the class imbalance issue alleviated. GAD-NR [22] aimed to reconstruct the entire neighborhood of a node, encompassing the local structure, self-attributes, and neighbor attributes and identify anomalous nodes with neighborhood reconstruction loss.

### 5.2 Federated Graph learning

FedAvg [16] introduced the concept of federated learning (FL), where a central server distributes a global model to multiple clients, who then train the model locally using their own data. Afterward, the clients upload their locally trained weights to the server, which aggregates these weights to update the global model. FedProx [11] extended FedAvg by introducing a regularization term that penalizes the deviation between client and global model parameters, stabilizing the learning process.

In federated graph learning, a unique challenge arises due to the absence of connections between nodes across different client subgraphs. FASTGNN [28] introduced an edge generator at the server, which reconstructed edges between clients using Gaussian-randomly generated edges to simulate inter-client connections. Fed-Sage [31], proposed a neighbor generator to predict the number of missing neighboring nodes and reconstruct their hold-out features. Building upon this, FedNI [20] improved the neighbor generation process by incorporating a discriminator that distinguishes between real missing neighbor features and those generated by the model. FedEgo [32] assumed that the edges between client subgraphs are known and employed Mixup [30] to create mashed ego graphs, offering a level of privacy protection during the aggregation process. Further advancing the field, FedGL [4] proposed using global pseudo-labels and a global pseudo-graph, which are distributed to clients. These pseudo-structures enrich the local training labels and enhance the graph structure, mitigating the effects of missing edges. Lastly, FedGTA [12] introduced a personalized model aggregation approach that leverages the mixed moments of neighboring node features and uses local smoothing confidence to weight the aggregation, enabling a more tailored model for each client while maintaining the integrity of the global model.

## 6 Conclusion

In this paper, we introduce the task of federated graph anomaly detection and analyzed its challenge of identifying graph anomalies without inter-client edges. To address this challenge, we proposed a federated graph anomaly detection method based on disentangled representation learning. Node features are disentangled into intrinsic features and subgraph style features from views of feature and structure to tackle above challenges. Contrastive learning strategies are then adopted to generate negative node pairs with CVAE, effectively addressing the class imbalance issue in anomaly detection and achieve feature disentanglement. Next, pseudo-features are generated for a few nodes within each client subgraph that are likely to have connections outside the subgraph. These nodes are finally shared globally to simulate inter-client connections. Comprehensive experiments conducted on six datasets validate the effectiveness of our method in federated graph anomaly detection.

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
