# OpenReview forum: "Federated Graph Anomaly Detection via Disentangled Representation Learning"
_ACM.org/TheWebConf/2025/Conference — WWW 2025 Oral_

### Official Review · Reviewer_G9Hm · 2024-11-17

**Novelty:** 5
**Technical Quality:** 5

**Review:**

The paper introduces a method for federated graph anomaly detection using disentangled representation learning. The authors propose separating node features into intrinsic and subgraph style components to detect anomalous nodes while preserving data privacy across clients. They also implement a federated approach that shares pseudo-nodes derived from subgraph style features to simulate cross-client connections. Experiments on six datasets are conducted to demonstrate the proposed method’s effectiveness, with a detailed time complexity analysis included.

Pros and Cons
Pros:
	1.	The paper addresses a relevant and timely topic in graph anomaly detection, aligning with current research trends.
	2.	The authors provide a comprehensive time complexity analysis.

Cons:
	1.	Presentation and Clarity: The paper has numerous issues with presentation, including inconsistent notation and symbol usage and vague description, making it difficult to follow and reducing overall readability.
	2.	Unclear Motivation for Design Choices: Several key design choices, such as the use of disentangled representation learning and autoencoders, lack a clear explanation of the underlying motivation. The authors often reference previous work without providing justification specific to this paper’s context.
	3.	Unconvincing Design Decisions: Certain design aspects, such as the method for selecting pseudo-nodes, are based on assumptions that lack sufficient support and may not be convincing in practical graph scenarios.
	4.	Insufficient Experimental Validation: The experimental section is inadequately detailed, with limited explanations of the anomaly injection method, a lack of comparison with relevant baseline methods, and metrics that are not ideal for anomaly detection. Additionally, some unexpected results are presented without sufficient clarification, and the disentangled learning hypothesis is not fully validated through experiments.

Detailed Concerns and Suggestions

1. Presentation and Readability Issues

The paper’s presentation has multiple issues that make it difficult to understand and follow. Inconsistent notation, ambiguous symbols, and vague explanations lead to a significant drop in readability and clarity. Here are the specific areas where improvements are needed:

- In the Preliminary section, the symbol ‘Z’ is introduced within the hypothesis of disentangled learning, but it does not appear in Figure 3. Additionally, Figure 3 includes arrows whose meaning is left unexplained.
- Symbols ‘N’ and ‘n’ are used without any introduction to their meanings in ‘Hypothesis of Disentangled Learning’ section. Furthermore, the symbol ‘E’ represents “environment” in one section and “edge set” in the Problem Definition section. Reusing the same symbol for different concepts creates confusion.
- The symbol for the number of clients varies between ‘M’ (used in the problem definition) and ‘N’ (used in the time complexity analysis table). Similarly, the notation for the number of nodes switches between uppercase ‘N’ in equations 9 and 10 and lowercase ‘n’ in the complexity analysis. This inconsistency makes it hard for readers to follow the flow of information accurately.
- In equation (5), the symbols \mu^Y and ‘Y’ are introduced without a clear explanation of their roles, especially why ‘Y’ would be included in an autoencoder setting. If represents labels, this should be clarified, as using labels in this context is unusual.
- Equation (7) uses the symbol ‘C’ in a way that is distinct from its meaning in equations (4) and (6), yet the same symbol is reused, leading to ambiguity.
- Additional issues include the index ‘i’ in the product notation in equation (7), which does not reappear in subsequent term, and the unexplained appearance of ‘m’ in equation (8).
- In equation (11), the term “conditional variational autoencoder” is mentioned, yet it is unclear what the conditional information entails. If the label  ‘Y’ is the conditional information, this should be explicitly stated in the notation, such as using ‘q(\cdot | Y)’ or ‘p(\cdot | Y)’ to make it clear.
- There are conflicting statements about the InfoNCE loss in relation to mutual information bounds. The text first claims that InfoNCE can estimate the upper bound of mutual information, followed by a contradictory statement that it estimates the lower bound. From the structure of equation (12).
- The symbol ‘i’ appears in equations (19) and (20) without an accompanying explanation, which is only provided much later in the text, making the equations hard to interpret.
- Figures have legends with font sizes too small to read at standard size.


2. Motivation and Justification for Model Design Choices

The paper proposes disentangling node features into intrinsic features and subgraph style features as a central component of the model, yet it fails to adequately justify this design choice. The authors do not explain why disentangled learning is particularly suited for graph anomaly detection or how it aligns with the specific challenges of federated graph learning.

Furthermore, the use of an autoencoder, specifically one with an MLP-based encoder, is mentioned with minimal explanation, justified only by a brief reference to prior work. Merely citing previous studies does not suffice. Each module’s purpose and relevance to the anomaly detection task should be clearly explained.


3. Questionable Assumptions in Pseudo-Node Selection

One of the key design choices in this paper involves selecting pseudo-nodes from each client based on their distance from the mean representation within the subgraph, under the assumption that nodes far from the mean are more likely to connect to other clients’ nodes. This assumption, however, lacks practical support and is unconvincing. Here’s a simple and common counterexample: imagine a social network with 10 nodes, from node 1 to node 10. Node 1 is connected to every other node (i.e., node 1 is the central node, a structure often seen in real-world graphs). Now, if we divide the network into two parts as subgraphs for two clients: nodes 1-5 as the first subgraph and nodes 6-10 as the second subgraph. Since node 1 is the central node in the first subgraph, the calculated mean will be very close to the representation of node 1. According to the criteria in the article, node 1 should theoretically be unlikely to connect with nodes from other clients. However, in reality, it’s precisely node 1 that is connected to nodes from other clients. This is a simple example to illustrate that the assumptions in the article are not entirely convincing and need further consideration. To substantiate this approach, the authors should provide a more detailed analysis, potentially by examining structural properties of whole-graph connections or by establishing conditions under which this assumption holds true.


4. Experimental Design and Evaluation Concerns

- Anomaly injection plays a central role in evaluating anomaly detection methods, yet the paper merely states that it “follows a previous work” without providing details. Given its significance, a comprehensive description of the anomaly injection process is essential.
- The baseline comparisons used in the paper consist of general federated learning methods and graph anomaly detection techniques. However, a quick google gives me a federated graph anomaly detections study titled “Federated Graph Anomaly Detection via Contrastive Self-Supervised Learning,” published in TNNLS journal in June 2024 (four months before the WWW’25 submission). This paper addresses the similar research question and would provide a highly relevant point of comparison.
- AUC is widely considered a more informative metric for graph anomaly tasks rather than ACC.
- The results in Table 3 show identical scores for FedAvg and FedProx across several datasets (e.g., Cora, Flickr, and BlogCatalog). Given the differences between these two models, this result is surprising and raises concerns about the implementation’s validity. A verification of the code implementation for these models would help ensure that the reported results are accurate.
- Although the paper emphasizes disentangled learning, the experimental section lacks sufficient validation for this approach. Section 2.1 proposes a hypothesis about the relationships between factors in disentangled learning, yet the experiments only briefly examine the separability of ‘C’ and ‘S’ with respect to the target label ‘Y’ in Section 4.5. Other factors and relationships outlined in the hypothesis are not explored, leaving the claim about disentangled learning’s role in this task unverified.

**Questions:**

1. Motivation for Disentangled Representation: Could the authors elaborate on why disentangled representation learning is particularly suited to graph anomaly detection within a federated learning context? How does it specifically address challenges inherent to this task?
2. Assumptions in Pseudo-Node Selection: The paper assumes that nodes far from the mean in each subgraph are more likely to connect to nodes in other clients. This assumption appears unconvincing, could the authors provide additional evidence or analysis to justify this assumption?
3. Baseline Comparisons: Considering the relevance of recent work on federated graph anomaly detection, such as the June 2024 TNNLS paper on contrastive self-supervised learning, why was this study not included as a baseline? How does the proposed method compare to this recent approach?

**Reviewer Confidence:**

4: The reviewer is certain that the evaluation is correct and very familiar with the relevant literature

**Scope:**

3: The work is somewhat relevant to the Web and to the track, and is of narrow interest to a sub-community

---

### Official Review · Reviewer_JtWd · 2024-11-21

**Novelty:** 6
**Technical Quality:** 6

**Review:**

This paper introduces a federated graph anomaly detection method that leverages disentangled representation learning, aiming to address the challenges in cross-client anomaly detection tasks. The method utilizes CVAE to extract intrinsic and subgraph-style features from node attributes and graph structure. By generating negative node pairs with CVAE, the method employs contrastive learning for feature disentanglement, which helps address class imbalance issues in anomaly detection. To further enhance performance, the authors construct a global shared node pool from the local feature spaces of each client, facilitating the training of local models based on this pool.

**An evaluation of the  quality:**

1. Compared to earlier approaches in federated learning for graphs, which were not tailored for anomaly detection, the proposed method delivers competitive results across multiple metrics.

2. The paper provides a comparative analysis of the time and space complexity of the method, demonstrating that it is comparable in efficiency to current federated learning approaches. Furthermore, since the model uses batch sampling for subgraphs during training and the number of global shared nodes is limited, the model is also scalable.

3. The paper includes a large number of experiments, listing the results and standard deviations, which show the robustness of the model.

**An evaluation of the  clarity:**

The quality of some figures could be improved for better clarity. For example, Figure 1 would benefit from a legend, and the font size of legends in other experimental figures could be increased to enhance readability.

**An evaluation of the  originality:**

1.Research in this domain is still limited, and this work offers valuable insights for future developments in federated anomaly detection on graph data.

2.The use of pseudo-node features for sharing ensures that the privacy of client data is maintained, addressing a key concern in federated learning applications.

**An evaluation of the significance:**

The method assumes no overlapping nodes between clients, as is common in previous federated graph learning approaches. However, in practical applications, graph data from different clients may have significant overlap in nodes, which could impact the effectiveness of the current approach. It would be beneficial for the authors to discuss how the method might be adapted to handle such scenarios, as well as outline the limitations of their approach in this context in the future work section.

**Questions:**

1.Could authors further elaborate on the dependencies between the model components designed to address the federated setting and the graph anomaly detection problem in this paper?

2.This paper adopts the traditional approach of partitioning subgraphs to simulate client subgraphs in federated graph learning. Could authors analyze whether the partition's quality of subgraphs has any impact on the performance of the model for this task?

3.Other questions can be referred to the comments in **Review** section.

**Reviewer Confidence:**

4: The reviewer is certain that the evaluation is correct and very familiar with the relevant literature

**Scope:**

4: The work is relevant to the Web and to the track, and is of broad interest to the community

---

### Official Review · Reviewer_6ZTT · 2024-12-01

**Novelty:** 5
**Technical Quality:** 4

**Review:**

This paper addresses the problem of anomalous node detection in federated scenarios by disentangling node features into intrinsic features and subgraph style features, aiming to address graph anomalies and missing inter-client edges. The paper designs an innovative method by constructing a global shared node pool and selecting a few nodes from it to solve the issue of missing connections.

Pros:
1. Significance: The paper designs an innovative method to solve the issue of missing connections in federated graph anomaly detection.
2. Originality: The method of disentangling intrinsic features and subgraph-style features in federated learning is a notable innovation. The pseudo-intrinsic features of shared nodes are generated using VAE, ensuring that the privacy of client node features is preserved.
3. Quality：The paper presents a large number of experiments, including results and standard deviations, which effectively demonstrate the robustness of the proposed model.
4. Clarity: The paper provides thorough theoretical analysis, uses appropriate datasets, and conducts comprehensive experiments to validate the efficacy of the proposed method.

Cons:
1. Clarity: The authors could consider structuring the theoretical part of the Methodology by presenting the conclusions and intuitive reasoning first, followed by the detailed reasoning process. This approach would help make the paper's structure clearer.
2. Significance：From the author's analysis, it can be inferred that the method has a comparable time overhead to typical federated graph learning approaches, though it slightly increases memory overhead.

**Questions:**

1. In Section 4.4, the appropriate hyperparameters were determined through experiments. Please analyze potential reasons behind this combination of values.
2. It is possible to maintain the performance of the detection model without increasing the space overhead on both the client and server sides?

**Reviewer Confidence:**

4: The reviewer is certain that the evaluation is correct and very familiar with the relevant literature

**Scope:**

4: The work is relevant to the Web and to the track, and is of broad interest to the community

---

### Official Review · Reviewer_59nv · 2024-12-02

**Novelty:** 6
**Technical Quality:** 6

**Review:**

This paper presents a novel and effective approach to anomalous node detection in cross-client scenarios, where graph data is partitioned across different clients for privacy reasons. The proposed method tackles the challenge of missing edges in the graph due to these partitions by extracting both intrinsic features and subgraph-style features from node characteristics and graph structure. By employing a Conditional Variational Autoencoder (CVAE) to generate negative node pairs and utilizing contrastive learning for feature disentangling, the approach effectively improves anomaly detection performance. Additionally, the paper introduces the concept of a global shared node pool to select representative nodes, which helps in mitigating the issue of missing connections across clients. Local models are trained based on the feature space S from each client, allowing the method to operate efficiently in cross-client settings.

Pros:
1. Significance: The paper focuses on a real-world problem of missing edges in graph anomaly detection tasks due to data partitioning for privacy, providing an innovative solution that achieves strong experimental results.
2. Originality: The proposed method of feature disentangling through contrastive learning is a promising technique with potential for use in a wide range of applications beyond the scope of this paper.
3. Quality and Clarity: The paper presents thorough theoretical analysis, employs suitable datasets, and conducts comprehensive experiments to validate the proposed method’s efficacy.

Cons:
1. Clarity: The paper introduces many symbols throughout the text, which could be overwhelming for readers. Including a table summarizing the key symbols would improve clarity and help readers navigate the content more easily.
2. Clarity: The Preliminaries section would benefit from a brief introduction to federated learning, especially for readers less familiar with this concept. Alternatively, the related work section could be moved earlier in the paper to provide necessary background information for a better understanding of the problem and methodology.

**Questions:**

1. Since using CVAE inevitably increases space overhead, could the authors analyze its necessity and reasons why it is required for the proposed method?
2. The problem setting in this paper follows FedSage, where there are no overlapping nodes across different clients. From a practical application perspective, how do overlapping nodes manifest in real-world scenarios? If overlapping nodes were to occur, could the proposed method be adapted or transformed in some simple way to handle such cases? What are the limitations of the approach in addressing this issue?

**Reviewer Confidence:**

4: The reviewer is certain that the evaluation is correct and very familiar with the relevant literature

**Scope:**

4: The work is relevant to the Web and to the track, and is of broad interest to the community